# Serum Bilirubin and Markers of Oxidative Stress and Inflammation in a Healthy Population and in Patients with Various Forms of Atherosclerosis

**DOI:** 10.3390/antiox11112118

**Published:** 2022-10-27

**Authors:** Libor Vítek, Alena Jirásková, Ivana Malíková, Gabriela Dostálová, Lenka Eremiášová, Vilém Danzig, Aleš Linhart, Martin Haluzík

**Affiliations:** 1Institute of Medical Biochemistry and Laboratory Diagnostics, 1st Faculty of Medicine, Charles University and General University Hospital in Prague, Kateřinská 32, 121 08 Prague, Czech Republic; 24th Department of Internal Medicine, 1st Faculty of Medicine, Charles University and General University Hospital in Prague, U. Nemocnice 2, 128 08 Prague, Czech Republic; 32nd Department of Internal Medicine, 1st Faculty of Medicine, Charles University and General University Hospital in Prague, U. Nemocnice 2, 128 08 Prague, Czech Republic; 4Diabetes Centre, Institute of Clinical and Experimental Medicine, Vídeňská 1958/9, 140 00 Prague, Czech Republic

**Keywords:** atherogenesis, atherosclerosis, bilirubin, inflammation, oxidative stress

## Abstract

Oxidative stress and inflammation contribute significantly to atherogenesis. We and others have demonstrated that mildly elevated serum bilirubin levels protect against coronary and peripheral atherosclerosis, most likely due to the antioxidant and anti-inflammatory activities of bilirubin. The aim of the present study was to assess serum bilirubin and the markers of oxidative stress and inflammation in both healthy subjects and patients with various forms of atherosclerosis. The study was performed in patients with premature myocardial infarction (n = 129), chronic ischemic heart disease (n = 43), peripheral artery disease (PAD, n = 69), and healthy subjects (n = 225). In all subjects, standard serum biochemistry, *UGT1A1* genotypes, total antioxidant status (TAS), and concentrations of various pro- and anti-inflammatory chemokines were determined. Compared to controls, all atherosclerotic groups had significantly lower serum bilirubin and TAS, while having much higher serum high-sensitivity C-reactive protein (hsCRP) and most of the analyzed proinflammatory cytokines (*p* < 0.05 for all comparisons). Surprisingly, the highest inflammation, and the lowest antioxidant status, together with the lowest serum bilirubin, was observed in PAD patients, and not in premature atherosclerosis. In conclusion, elevated serum bilirubin is positively correlated with TAS, and negatively related to inflammatory markers. Compared to healthy subjects, patients with atherosclerosis have a much higher degree of oxidative stress and inflammation.

## 1. Introduction

During the recent decades, the negative relationship between atherosclerotic diseases and the concentration of serum bilirubin has repeatedly been confirmed (for a review, see [1]). Mildly elevated serum bilirubin levels in the absence of any underlying liver disease protect against coronary heart disease (CHD) [2] and peripheral artery disease (PAD), prolonging the time from the development of clinically important carotid atherosclerosis by almost 20 years [3]. In fact, serum bilirubin below 7 μmol/L increases the risk of CHD by 30% [4], and serum bilirubin has been shown to have a predictive value for CHD as HDL cholesterol levels [5].

Several mechanisms may explain these observations. Bilirubin, which is considered the most potent endogenous antioxidant [6], ameliorates the high oxidative stress involved in atherogenesis [7]. Furthermore, bilirubin has potent immunosuppressive activities in almost all stages of the immune system [8], with the potential to inhibit the low-grade inflammation present in patients with accelerated atherogenesis [9]. In fact, inflammation is deeply involved in the process of atheromatous lipid accumulation [10], as evidenced by the neutrophilic myeloperoxidase oxidation of LDLs [11], which induces the production of proinflammatory mediators [12,13]. Inflammation is also promoted by other factors frequently present in patients with atherosclerosis. As an example, dyslipidemia activates the inflammatory functions of vascular endothelial cells [14]. Additionally, the same positive association with proinflammatory status is also known for arterial hypertension (via angiotensin II) [9], diabetes (via generation of proinflammatory advanced glycation end products) [9], or increased adiposity (fat tissue is an important source of proinflammatory cytokines) [15]. Hence, atherosclerosis can be seen as an immunometabolic disease involving chronic inflammation, oxidative stress, and metabolic dysfunction [16]. The atheroprotective effects of bilirubin are very versatile [1], involving antioxidative and anti-inflammatory activities, which are possibly amplified by its cell signaling activities [17], since bilirubin acts as a ligand for several cell receptors exerting real endocrine activities with beneficial metabolic effects [17,18].

Therefore, the objective of our study was to assess the relationship between serum bilirubin concentrations, the (TA)n *UGT1A1* gene promoter variants (being the most important genetic determinant of serum bilirubin concentrations) [6], and the markers of oxidative stress and inflammation in patients with various forms of atherosclerosis, as well as to assess whether these factors are more deteriorated in patients with premature atherosclerosis.

## 2. Materials and Methods

### 2.1. Subjects

The atherosclerotic patients were recruited from inpatients and outpatients of the Second Department of Internal Medicine, General University Hospital in Prague who were examined between 2012 and 2014. The study was carried out on 129 patients with premature coronary atherosclerosis, which was defined as the manifestation of myocardial infarction (MI), aged <45 for men and <55 for women, with examination performed 2–4 years after MI. In most patients with premature coronary atherosclerosis, no general atherosclerosis occurred, only with coronary artery disease present (commonly a single vessel disease). Additionally, 43 patients with chronic ischemic heart disease (IHD) and 69 patients with PAD were also included, for whom their diagnoses were made based on routine clinical cardiology and angiology examinations. For the control group, 225 healthy subjects were used for the comparisons of laboratory parameters.

The patient and control groups were divided into quartiles according to serum bilirubin concentrations, and the relationships between serum bilirubin and the analyzed markers were determined in each of the individual bilirubin quartiles.

Written informed consent was obtained from each patient included in the study, and the study protocol conformed to the ethical guidelines of the 1975 Declaration of Helsinki as reflected in a priori approval by the Ethics Committee of the General University Hospital in Prague.

### 2.2. Laboratory Analyses

In all subjects, standard serum biochemistry was determined by routine assays on an automated analyzer (Cobas R8000 Modular analyzer, Roche Diagnostics GmbH, Mannheim, Germany). The total serum antioxidant status (TAS) was determined spectrophotometrically (Randox, GB). In this assay, 2,2′-Azino-di-[3-ethylbenzthiazoline sulphonate] (ABTS) was incubated with a peroxidase (metmyoglobin) and H_2_O_2_ to produce the radical cation ABTS*^+^. This had a relatively stable blue-green color, which was measured at 600 nm. Antioxidants in the added sample caused a suppression of this color production to a degree which was proportional to their concentrations.

Serum high-sensitivity C-reactive protein (hsCRP) was determined by immunonephelometry (Behring Nephelometer II). The principle of the method was based on polystyrene particles coated with monoclonal antibodies to CRP that agglutinate when mixed with samples containing CRP. The intensity of the scattered light in the nephelometer depended on the CRP concentration present in the sample. For the hsCRP assay, we used an initial sample dilution of 1:20. The samples were automatically diluted 1:100, 1:400, and 1:2000. This assay could measure CRP concentrations from 0.175 mg/L to 1100 mg/L.

Serum cytokines and chemokines were immunochemically determined using a multiplex kit (Luminex, Linco Res., St Charles, MO, USA) working on xMAP bead-based technology to simultaneously detect and quantify multiple secreted proteins, including cytokines, chemokines, and growth factors.

The dinucleotide variations (TA)n (dbSNP rs81753472) in the *UGT1A1* gene promoter were determined in a subset of subjects in whom gDNA was available (patients with premature atherosclerosis, n = 75; IHD, n = 35; PAD, n = 67; controls, n = 190). Analyses of these gene mutations were performed by fragment analysis using an automated capillary DNA sequencer as previously described [19].

### 2.3. Statistical Analyses

The data are expressed as mean ± SD, or as median and IQ range when the data were nonnormally distributed. The percentage counts were compared using the Chi-square test. ANOVA on Ranks with Dunn’s post hoc testing was used to compare variables among individual patient groups. The relationships between some of the variables analyzed were also evaluated using linear regression analyses. Logistic regression analysis was used to assess the predictive role of the variables analyzed. All analyses were performed with the alpha set to 0.05. The statistics were calculated using SigmaPlot v. 14.5 (Systat Software, Inc., San Jose, CA, USA).

## 3. Results

### 3.1. Serum Bilirubin Concentrations in Patients with Various Forms of Atherosclerosis

Compared to the controls, all groups of atherosclerotic patients had significantly lower serum bilirubin concentrations. This association was more pronounced in men compared to women (Table 1, Figure 1). Interestingly, compared to other atherosclerotic patients, the lowest serum bilirubin concentrations were observed in PAD patients (Table 1, Figure 1).

When serum bilirubin was analyzed according to the *UGT1A1* genotype, substantially lower bilirubin concentrations were observed across all genotypes in all groups of atherosclerotic patients (Table 2.). The differences in serum bilirubin were more pronounced in men, while they were less evident in women (Table 2).

Compared to controls, each micromolar increase in serum bilirubin concentration was associated with a significant decrease in the chances of presenting atherosclerotic diseases (Table 3). This negative association was again more pronounced in men and in patients with PAD (each micromolar increase in serum bilirubin concentration was associated with a 14% decrease in PAD risk).

The presence of the TA_7_ allele, responsible for higher serum bilirubin levels [6], was associated with a significantly decreased probability (approximately 65%) of IHD and PAD manifestation in males; a similar but nonsignificant trend was also observed in females and premature atherosclerosis (Table 4).

### 3.2. Relationship between Serum Bilirubin Concentration and TAS in Healthy Controls and Patients with Various Forms of Atherosclerosis

In healthy controls, subjects with the lowest serum bilirubin (1st quartile) had significantly lower serum TAS compared to subjects in the highest bilirubin quartile (*p* < 0.001), as well as subjects in the third (*p* = 0.001) and even in the second bilirubin quartile (*p* = 0.02) (Table 5a); this positive association was also confirmed in the linear regression analysis (Figure 2). Compared to the controls, all groups of atherosclerotic patients had significantly lower TAS (*p* < 0.05 for all comparisons between cases and controls) (Table 5b).

### 3.3. Relationship between Serum Bilirubin, hsCRP, and Other Inflammatory Markers

Subjects with the lowest serum bilirubin (first quartile) had significantly higher serum levels of hsCRP compared to subjects in the highest bilirubin quartile, as well as compared to the third bilirubin quartile (*p* = 0.01 for both comparisons, Table 6).

The observed relationship was also confirmed by linear regression analysis (Figure 3).

A similar association was also observed in patients with various forms of atherosclerosis, with the highest hsCRP concentrations found in patients with PAD (Table 7), who also had significantly higher serum concentrations of the proinflammatory cytokines TNF-α, IL-1β, IL-6, and IL-8, as well as other atherogenic factors such as VEGF, E-selectin, or ICAM (*p* < 0.05 for most of the comparisons between cases and controls, Table 7). On the other hand, serum concentrations of anti-inflammatory IL-10 were significantly lower in all atherosclerotic groups compared to controls (Table 7).

When considering the other risk factors that may account for the manifestation of atherosclerotic diseases, significantly higher BMI, as well as serum concentrations of blood glucose, triacylglycerols, and lower HDL cholesterol, were detected in patients with atherosclerosis (Table 8). These patients were older, with much higher smoking rates compared to the controls. On the other hand, most of these factors have in previous studies been reported to be associated with lower bilirubin concentrations [20].

## 4. Discussion

Our study confirms that mild elevation of serum bilirubin, even within the currently used physiological range, is associated with an increased antioxidant capacity and a decreased inflammatory status in healthy subjects, and further confirms that the same association was found in atherosclerotic patients. Accordingly, increased TAS was also reported in subjects with Gilbert syndrome (benign hyperbilirubinemia) in Australian studies [21] and one Turkish study [22]; a positive association between serum bilirubin and antioxidant capacity has also been reported in other clinical studies [23,24,25], even in newborns suffering from neonatal jaundice [26,27,28]. These data are consistent with the observation made in our previous study, in which a positive association between serum bilirubin and peroxyl radical scavenging activity was clearly demonstrated in healthy Czech and Italian subjects [24]. A synergistic interaction between water- and fat-soluble antioxidants is believed to amplify total antioxidant capacity [29], which, together with the biliverdin reductase-mediated bilirubin antioxidant amplification cycle [30], might account for the higher than stochiometric antioxidant capacity of bilirubin [23]. In in vitro experiments of this study, we were able to demonstrate a clear positive association between TAS and gradually increasing concentrations of bilirubin in the commercial serum with defined TAS [23]. The potent antioxidant effects of bilirubin observed in these studies agree with a significant negative association of bilirubin with plasma concentrations of oxysterols [24] and oxidatively modified LDLs [31,32,33], which significantly contribute to the progression of atherogenesis [34,35]. In fact, this was confirmed in numerous studies, in which coronary and peripheral atherosclerotic diseases were negatively associated with serum bilirubin concentrations [1]. Nevertheless, it should be noted that the TAS assay used in our study does not always completely reflect the oxidative stress status. The antioxidant defense system is very complex, and the use of a single method to analyze the presence of antioxidants in biological material, such as TAS, may be insensitive to subtle changes that can cause imbalance of the oxidation–antioxidant system. Thus, it would be more accurate to assess both antioxidant and oxidant status simultaneously with calculation of specific oxidative stress indexes to assess a dynamic equilibrium between oxidants and antioxidants in serum [36,37,38].

A negative association with serum bilirubin was also found for hsCRP, considered an unfavorable prognostic factor for atherosclerotic diseases [36]. In fact, an almost doubled risk of CHD was demonstrated in patients in the upper vs. lower hsCRP tertile in a study by Danesh et al. [39]. Based on these data, risk prediction algorithms have been developed using a simple quintile approach to the evaluation of hsCRP to improve global risk assessment in the primary prevention of cardiovascular disease [40]. In line with these facts, hsCRP has been found to be a strong predictor of NAFLD [41,42,43]; additionally, it has also proven to be a strong predictor of a metabolic syndrome, insulin resistance [41], diabetes [44,45], hypertension [46], and even all-cause mortalities [47]. It is interesting to note that serum bilirubin is negatively associated with all of these metabolic diseases, including NAFLD [48] or diabetes (for a review, see [49]).

Our observation of the negative association between serum bilirubin and hsCRP is consistent with previous reports. In fact, a similar correlation was also found in three large Korean studies conducted in healthy subjects [50,51,52], as well as in an Italian study on outpatients with normal liver function tests and no apparent inflammatory disease [53]. Moreover, a similar negative association was found in patients with metabolic syndrome [54,55,56], hypothyroidism [57], being overweight, insulin resistance [58], obstructive sleep apnea [59], or even migraines [60].

Similar data on the suppressive effects of bilirubin for proinflammatory cytokines have also been reported. In fact, bilirubin was found to decrease the levels of inflammatory cytokines, as has been reported in experimental studies [61,62,63]; this is an observation consistent with the general immunosuppressive activities of bilirubin at almost all stages of the immune system, including innate as well as adaptive immunity [8].

It is important to note that per se CRP is a potent modulator of pro- and anti-inflammatory cytokines, as well as other chemokines and adhesion molecules that are also implicated in the process of atherosclerosis [64]. CRP, an evolutionary conserved plasma protein of the innate immune system, is integrally involved in the systemic response to inflammation. CRP activates complement, stimulates phagocytosis, and binds to immunoglobulin receptors (FcγR) [65]. CRP synthesis is induced in liver cells by acute phase stimuli, particularly proinflammatory cytokines IL-6 and IL-1β [65]. Our data are in accord with this, showing much lower concentrations of both proinflammatory cytokines and hsCRP. Interestingly, the highest values were observed in PAD patients (Table 7). CRP also modulates the expression of the anti-inflammatory cytokine IL-10 that plays a protective role in atherogenesis [66,67]. Its concentrations showed the opposite trend, being highest in the healthy controls and lowest in the PAD patients (Table 7). We have also shown that patients with premature atherosclerosis had significantly higher concentrations of VEGF-A (Table 7), which is also induced by CRP [68].

Similar associations with serum bilirubin for E-selectin and ICAM-1 were found (Table 7). P- and E-selectins, C-type lectins that bind sialylated and fucosylated carbohydrate ligands, are expressed in both acutely and chronically inflamed endothelium; further, they serve as rolling molecules for monocytes, neutrophils, effector T cells, B cells, and natural killer cells [69]. Together, P- and E-selectins play an important role in both the early and advanced stages of atherosclerotic lesions, since P- and E-selectins are adhesion molecules that mediate the first step in the extravasation of leukocytes [70]. CRP has been shown to induce the expression of E-selectin in atherosclerotic plaques [67] and P-selectin in platelets [68]. ICAM-1, a member of the immunoglobulin superfamily, is also involved in atherogenesis, presumably through the regulation of monocyte recruitment into areas prone to atherosclerosis. ICAM-1 expression is elevated in atherosclerosis-prone aortas and is regulated by proinflammatory stimuli. CRP, as well as oxLDL, induce endothelial cell ICAM-1 expression [65,71].

Collectively, we found important associations between serum bilirubin and these chemokines, suggesting the complex contribution of bilirubin to those immune system factors implicated in atherogenesis.

Surprisingly, the associations between bilirubin concentrations in patients with atherosclerosis were most pronounced in PAD, compared to IHD and premature IM (Table 7). The explanation could lie in the fact that our premature atherosclerotic patients, almost exclusively with coronary artery disease that often only affected a single vessel, had unstable atherosclerotic plaques that accounted for the severity and prematurity of CHD. Meanwhile, in the other atherosclerotic groups, the pathologic process was long-lasting and affected multiple vessels. The presence of the TA_7_ allele, responsible for higher serum bilirubin levels [6], was associated with a significantly decreased probability (approximately 65%) of IHD and PAD manifestation in males, with a similar but nonsignificant trend also observed in females and premature atherosclerosis. This is not surprising, since Gilbert syndrome, which is caused by TA_7_ allele insertion in the *UGT1A1* promoter region and characterized with mild unconjugated hyperbilirubinemia, is associated with low prevalence as well as incidence of ischemic heart disease [20]. The weaker association between serum bilirubin and examined outcomes seems to be due to lower physiological serum bilirubin levels in women, together with the presence of the other factors protecting from or delaying atherogenesis which are present in women (such as production of estrogens) [6].

Patients with atherosclerosis were found to have important additional risk factors for coronary and peripheral heart disease (BMI, age, smoking status, dyslipidemia, and impaired glucose metabolism); significantly, all of these factors are also well known to be associated with lower bilirubin concentrations [20].

The study has some limitations, such as the fact that only TAS assay was used to assess the antioxidant status of the subjects examined, and also that the sample size of our study was not large enough to use a multivariate statistical approach.

## 5. Conclusions

Serum bilirubin is closely associated with markers of oxidative stress and inflammation, both in healthy controls and in patients with atherosclerosis. The manifestation of atherosclerotic diseases is associated with low serum bilirubin levels, especially in men. These findings support previously proposed calls for the reassessment of current decision limits of serum/plasma bilirubin concentrations [72,73]. Compared to healthy controls, patients with atherosclerosis are exposed to much higher levels of oxidative and inflammatory stresses. However, patients with premature atherosclerosis do not differ substantially from patients with other forms of atherosclerosis. On the contrary, they have much less pronounced elevations of oxidative and inflammatory stress markers, suggesting the involvement of etiopathogenic factors other than increased oxidative stress, which in these patients must be involved in the pathogenesis of premature coronary atherosclerosis.

## Figures and Tables

**Figure 1 antioxidants-11-02118-f001:**
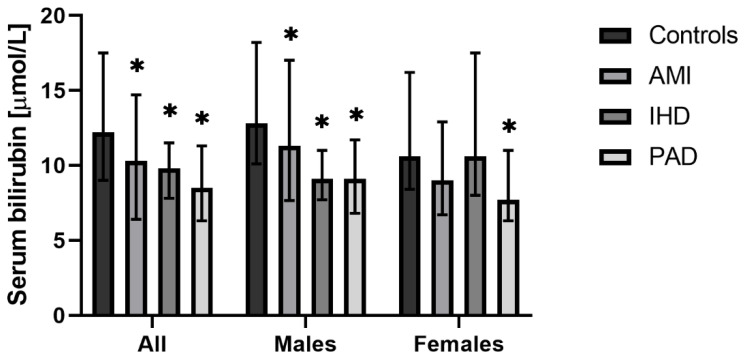
Serum bilirubin concentrations in patients with various forms of atherosclerosis. AMI, premature acute myocardial infarction; IHD, ischemic heart disease; PAD, peripheral artery disease. * *p* < 0.05, when compared to controls.

**Figure 2 antioxidants-11-02118-f002:**
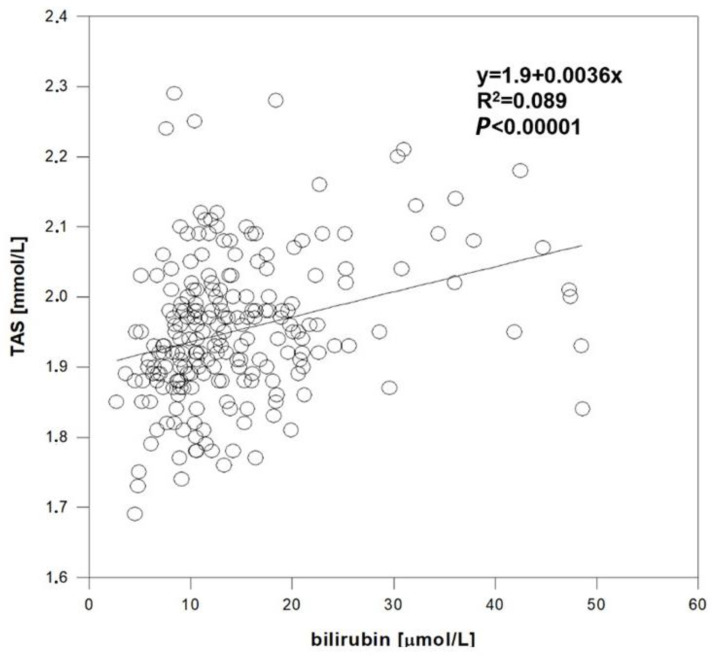
Relationship between serum bilirubin concentration and TAS in healthy controls.

**Figure 3 antioxidants-11-02118-f003:**
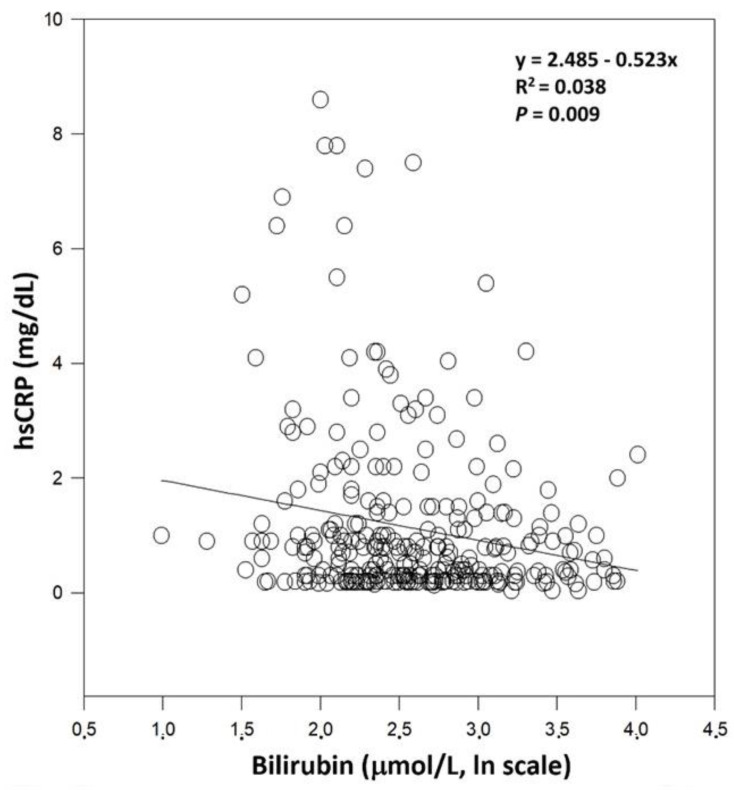
Relationship between serum bilirubin concentration and hsCRP in healthy controls.

**Table 1 antioxidants-11-02118-t001:** Serum bilirubin concentrations in patients with various forms of atherosclerosis.

Group	Premature Atherosclerosis	IHD	PAD	Controls
	(n = 129)	(n = 43)	(n = 69)	(n = 225)
All	10.3 *	9.8 *	8.5 *	12.2
	[7.6–14.7]	[7.8–11.5]	[6.3–11.3]	[9.0–17.5]
Males	11.3 *	9.1 *	9.1 *	12.8
	[7.7–16.2]	[7.7–11.0]	[6.8–11.7]	[10.1–18.2]
Females	9.0	10.6	7.7 *	10.6
	[6.7–12.9]	[8.0–17.5]	[6.3–11]	[8.4–16.2]

Serum bilirubin expressed in μmol/L; data expressed as median [25–75%]. * *p* < 0.05, when compared to controls. ANOVA on Ranks with Dunn’s post hoc analyses were used for all pairwise comparisons. IHD, ischemic heart disease; PAD, peripheral artery disease.

**Table 2 antioxidants-11-02118-t002:** Serum bilirubin concentrations in patients with atherosclerosis according to the *UGT1A1* genotype.

All	Premature Atherosclerosis	IHD	PAD	Controls
	(n = 75)	(n = 35)	(n = 67)	(n = 190)
6/6	8.2 *	8.2 *	7.6 *	10.1
	[6.8–11.2]	[6.8–10.5]	[6.3–11]	[7.9–12.3]
6/7	12	10	8.2 *	12.7
	[7.9–16.2]	[7.6–12]	[5.6–10.1]	[9.2–16.2]
7/7	18.8	25.3	12.2 *	22.3
	[10.5–25.7]	[14.9–36.1]	[8.8–23]	[17.7–35.2]
Males	(n = 58)	(n = 23)	(n = 34)	(n = 103)
6/6	8.3 *	8.5 *	8.7	10.6
	[6.8–11.9]	[6.2–10.7]	[6.3–12.5]	[9.0–12.7]
6/7	12.6	9.9 *	9.1 *	14.2
	[10.3–16.8]	[8.0–11.7]	[7.3–10.5]	[10.6–18.6]
7/7	18.8 *	ND	17.9	29.7
	[9.3–25.1]		[6.2–31.2]	[17.7–42.6]
Females	(n = 17)	(n = 12)	(n = 33)	(n = 87)
6/6	7.9	8.1	7.1	9.1
	[6.4–10.4]	[6.2–9.9]	[6.2–9.8]	[6.9–12.2]
6/7	7.5	10.6	7.7 *	10.4
	[5.7–10.8]	[8.4–14.2]	[5.1–9.4]	[8.4–14.2]
7/7	ND	26.2	12.2 *	20.8
		[11.7–39.4]	[9.0–16.1]	[17.5–25.2]

Serum bilirubin expressed in μmol/L, data given as median and 25–75%, * *p* < 0.05, when compared to controls. ANOVA on Ranks with Dunn’s post hoc analyses were used for all pairwise comparisons. ND, not detected; IHD, ischemic heart disease; PAD, peripheral artery disease. 6/6 (wildtype); 6/7 and 7/7 (associated with Gilbert syndrome) denotes number of TA repeats in the *UGT1A1* gene TATA box promoter (*UGT1A1* [TA]_n_).

**Table 3 antioxidants-11-02118-t003:** Risk of atherosclerotic disease with each μmol/L increase of serum bilirubin concentration.

Group	Premature Atherosclerosis	IHD	PAD
	(n = 129)	(n = 43)	(n = 69)
All	0.98	0.93 *	0.86 *
	[0.95–1.01]	[0.87–0.99]	[0.81–0.92]
Males	0.94 *	0.82 *	0.86 *
	[0.57–0.99]	[0.72–0.93]	[0.79–0.95]
Females	0.97	1.02	0.86 *
	[0.90–1.05]	[0.95–1.10]	[0.78–0.95]

Logistic regression analysis was used to assess the data, which are expressed as OR and 95% CI. IHD, ischemic heart disease; PAD, peripheral artery disease. * *p* < 0.05.

**Table 4 antioxidants-11-02118-t004:** Risk of atherosclerotic disease according to *UGT1A1* [TA]_7_ status.

Group	Premature Atherosclerosis	IHD	PAD
	(n = 75)	(n = 35)	(n = 67)
All	0.48 *	0.49 *	0.56 *
	[0.29–0.82]	[0.25–0.96]	[0.32–0.98]
Males	0.57	0.35 *	0.35 *
	[0.30–1.11]	[0.14–0.88]	[0.16–0.78]
Females	0.39	0.83	0.75
	[0.13–1.12]	[0.22–3.2]	[0.33–1.70]

Logistic regression analysis was used to assess frequency of the TA_7_ allele responsible for higher serum bilirubin levels, with odds for atherosclerosis manifestation. Data represent OR and 95% CI. IHD, ischemic heart disease; PAD, peripheral artery disease. * *p* < 0.05.

**Table 5 antioxidants-11-02118-t005:** (**a**) Relationship between serum bilirubin concentration and TAS in healthy controls. (**b**) Serum TAS in patients with various forms of atherosclerosis.

(a)				
	Bilirubin Quartile 1	Bilirubin Quartile 2	Bilirubin Quartile 3	Bilirubin Quartile 4
Bilirubin (μmol/L, min-max)	2.7–9	9.1–12.5	12.6–18.6	18.9–55.2
TAS (mmol/L, median [25–75%)	1.9 [1.87–1.95]	1.95 [1.89–2.01]	1.97 [1.90–2.01]	1.98 [1.92–2.06]
*p*-value		0.02	0.001	<0.0001
(b)				
	Premature Atherosclerosis (n = 129)	IHD (n = 43)	PAD (n = 69)	Controls (n = 225)
TAS (mmol/L, median [25–75%])	1.34 * [1.24–1.43]	1.39 * [1.29–1.48]	1.36 * [1.25–1.5]	1.94 [1.89–2.01]

ANOVA on Ranks with Dunn’s post hoc analyses were used for all pairwise comparisons. * = *p* < 0.05. IHD, ischemic heart disease; PAD, peripheral artery disease; TAS, total antioxidant status.

**Table 6 antioxidants-11-02118-t006:** Relationship between serum bilirubin concentration and hsCRP in healthy controls.

	Bilirubin Quartile 1	Bilirubin Quartile 2	Bilirubin Quartile 3	Bilirubin Quartile 4
bilirubin (μmol/L, min-max)	2.7–9.0	9.1–12.5	12.6–18.6	18.9–55.2
hsCRP (mg/L, median [25–75%])	0.9 (0.30–2.18)	0.7 (0.23–1.35)	0.45 (0.20–1.05)	0.44 (0.20–1.18)
*p*-value		NS	0.012	0.01

ANOVA on Ranks with Dunn’s post hoc analyses were used for all pairwise comparisons.

**Table 7 antioxidants-11-02118-t007:** Relationship between serum bilirubin concentration and hsCRP and proinflammatory cytokines in patients with atherosclerosis.

	Premature Atherosclerosis	IHD	PAD	Controls
	(n = 129)	(n = 43)	(n = 69)	(n = 225)
hsCRP	1.0 *	1.3 *	2.1 *	0.45
[mg/L]	[0.4–2.25]	[0.8–3.3]	[1.05–4.6]	[0.2–1.0]
TNF-α	6.9 *	7.3 *	7.6 *	4.4
[ng/L]	[4.7–8.5]	[5.8–9.1]	[5.3–10.8]	[3.2–6.1]
IL-1β	5.7 *	9.2 *	9.5 *	0.3
[ng/L]	[3.5–18.3]	[3.2–17.4]	[5.7–21.7]	[0.13–4.3]
IL-6	3.2 *	4.2 *	5.1 *	0.96
[ng/L]	[1.6–9.1]	[2.1–6.9]	[3.2–8.7]	[0.48–2.9]
IL-8	14.1 *	10.6	13.3*	7.9
[ng/L]	[9.6–22.9]	[8–16.7]	[12.1–34.7]	[5.5–11.1]
IL-10	0.29 *	0.42 *	0.25 *	0.98
[ng/L]	[0.2–0.4]	[0.3–0.5]	[0.2–0.3]	[0.3–1.6]
VEGF-A	413 *	361	ND	234
[ng/L]	[272–605]	[171–573]		[170–450]
*p*-selectin	96	80	90	85
[ng/L]	[75–113]	[66–97]	[64–106]	[70–102]
E-selectin	40 *	33	33	29
[ng/L]	[32–47]	[23–41]	[27–43]	[23–37]
ICAM	282 *	258 *	314 *	211
[ng/L]	[237–371]	[213–298]	[257–380]	[180–244]

Data expressed as median [25–75%]; ANOVA on Ranks with Dunn’s post hoc analyses were used for all pairwise comparisons. * *p* < 0.05. IHD, ischemic heart disease; PAD, peripheral artery disease; TAS, total antioxidant status; ND, not done.

**Table 8 antioxidants-11-02118-t008:** Metabolic parameters and other factors that affect the development of atherosclerosis.

	Premature Atherosclerosis	IHD	PAD	Controls
	(n = 129)	(n = 43)	(n = 69)	(n = 225)
Age	44 *	72 *	62 *	39
(years)	[40–47]	[68–80]	[49–75]	[30–6]
Sex (M:F, %)	76 *	70 *	51	48
Smoking (%)	89 *	52 *	88 *	15
BMI	27.6 *	26.9 *	27.4 *	24.2 *
(kg/m^2^)	[24.5–29.8]	[24.9–29.8]	[24.5–28.7]	[22.4–25.4]
Glucose	5.1 *	5.8 *	5.8 *	4.7
(mmol/L)	[4.7–5.5]	[5.3–6.8]	[5.3–6.6]	[4.8–5]
Total cholesterol	4.23 *	4.35 *	4.75	5.13
(mmol/L)	[3.5–4.9]	[3.8–5]	[4.1–5.7]	[4.4–5.7]
LDL cholesterol	2.22 *	2.45	2.7	2.77
(mmol/L)	[1.7–2.8]	[1.9–3.1]	[2–3.5]	[2.4–3.3]
HDL cholesterol	1.1 *	1.36 *	1.36	1.6
(mmol/L)	[1–1.3]	[1.1–1.6]	[1.2–1.6]	[1.3–1.9]
Triacylglyceroles	1.71 *	1.17	1.29	1.13
(mmol/L)	[1.1–2.7]	[1–1.5]	[0.9–1.8]	[0.8–1.5]

Percentage counts were compared using Chi-square testing. Other data are expressed as median [25–75%]. ANOVA on Ranks with Dunn’s post hoc analyses were used for all pairwise comparisons. * *p* < 0.05. All patients were treated with statins and platelet antiaggregation therapy. IHD, ischemic heart disease; PAD, peripheral artery disease.

## Data Availability

The data are contained within the article.

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
