# Peer review of "Serum Bilirubin and Markers of Oxidative Stress and Inflammation in a Healthy Population and in Patients with Various Forms of Atherosclerosis"

_antioxidants, 2022, doi:10.3390/antiox11112118_

Round 1
Reviewer 1 Report
In the present manuscript, Libor Vítek et al. try to correlate makers of inflammation and oxidative stress (Total serum antioxidant status, TAS) with bilirubin levels in patients with atherosclerosis. While the manuscript is well-written and many patients are included, major flaws significantly limit the quality of the study and leads to an overstated argument. A major concern is the use of a single unspecific assay to evaluate patient oxidative stress, which does not necessarily associate with bilirubin antioxidant capacity. For this aim, the TAS assay was chosen, which has well-known confounders and is not acceptable for current standards in the field. Down this line, material and methods description is highly vague. In addition, multivariate statistical analysis has not been considered.
Minor:
The relevance of measuring (TA)n in the UGT1A1 is not properly introduced in the introduction section, and obtained findings are not discussed in the discussion section.
The article does not seem to sufficiently fit the scope of the special issue.
Author Response
Pont-by-point reply to reviewers´ comments
First, we would like to thank the reviewers for careful review of our paper and insightful evaluation. We feel that their comments which we tried to fully address, substantially improved our Ms. For your convenience, all changes in the Ms. are highlighted by yellow color.
Reviewer 1
In the present manuscript, Libor Vítek et al. try to correlate makers of inflammation and oxidative stress (Total serum antioxidant status, TAS) with bilirubin levels in patients with atherosclerosis. While the manuscript is well-written and many patients are included, major flaws significantly limit the quality of the study and leads to an overstated argument.
Major comments
- A major concern is the use of a single unspecific assay to evaluate patient oxidative stress, which does not necessarily associate with bilirubin antioxidant capacity. For this aim, the TAS assay was chosen, which has well-known confounders and is not acceptable for current standards in the field.
Reply: Thank you for this comment. We are well aware of nonspecificity of the TAS assay used in our study. We used this assay since our previous in vitro results with commercial serum with defined TAS (Randox Lab.) published in 2002 demonstrated a clear positive association between TAS and gradually increasing concentrations of bilirubin (see Vitek et al. Gilbert syndrome and ischemic heart disease: a protective effect of elevated bilirubin levels. Atherosclerosis 2002. See also our Reply to Comment 2 of Reviewer 3).
- Down this line, material and methods description is highly vague.
Reply: The MM section was expanded to provide more details on the methods used in our studies (see p. 3).
- In addition, multivariate statistical analysis has not been considered.
Reply: Thank you for this comment. Based on sample size analyses, multivariate statistical analysis has not been considered as reliable.
Minor comments:
- The relevance of measuring (TA)n in the UGT1A1 is not properly introduced in the introduction section, and obtained findings are not discussed in the discussion section.
Reply: Information on (TA)n UGT1A1 has been added into the Introduction Section (see p. 2), and properly discussed in the Discussion Section (see p. 13).
Reviewer 2 Report
Dear Authors, In my view this article is well presented. I suggest that you should provide some graphic including graphical abstract. Also tables please present with more interesting image. Plese present results in graphs too. Sincerely
Author Response
Pont-by-point reply to reviewers´ comments
First, we would like to thank the reviewers for careful review of our paper and insightful evaluation. We feel that their comments which we tried to fully address, substantially improved our Ms. For your convenience, all changes in the Ms. are highlighted by yellow color.
In my view this article is well presented.
- I suggest that you should provide some graphic including graphical abstract.
Reply: As recommended, a graphical abstract has been added.
- Also tables please present with more interesting image. Please present results in graphs too.
Reply: Graph for Tab. 1 was added (see p. 4), Tabs. 5-6 have their graphic expression in Figs. 2-3 (originally 1 and 2) depicting linear regression analyses.
Reviewer 3 Report
The authors have performed a correlation study of the association of bilirubin levels with markers of inflammation in patients with various vascular diseases. The find that lower bilirubin levels are generally associated with increased risk of vascular disease. The topic is of interest and the data supportive. There are some weaknesses that need to be addressed.
1. The bilirubin levels reported are within normal limits for all patients. This raises the question of what role bilirubin plays. Is low bilirubin a cause of increased inflammation or just another marker? In the TAS measurement, how much of the antioxidant capacity is from bilirubin? If substantial, this could explain the correlation. This needs to be discussed, Also, please provide more in formation on exactly what their TAS assay kit measures.
2. Please provide post hoc test used for all ANOVAs.
3, Please discuss possible reasons for differences in males vs females.
4. The discussion focusses largely on CRP with relatively little on mechanisms related to bilirubin. Especially for a special topic on heme oxygenase, more focus on bilirubin metabolism is in order.
Author Response
Pont-by-point reply to reviewers´ comments
First, we would like to thank the reviewers for careful review of our paper and insightful evaluation. We feel that their comments which we tried to fully address, substantially improved our Ms. For your convenience, all changes in the Ms. are highlighted by yellow color.
The authors have performed a correlation study of the association of bilirubin levels with markers of inflammation in patients with various vascular diseases. They find that lower bilirubin levels are generally associated with increased risk of vascular disease. The topic is of interest and the data supportive. There are some weaknesses that need to be addressed.
- The bilirubin levels reported are within normal limits for all patients. This raises the question of what role bilirubin plays. Is low bilirubin a cause of increased inflammation or just another marker?
Reply: Thanks for this important comment. In fact, serum bilirubin levels are within the normal limit – which, however, does not mean that they are in order. Lower serum/plasma concentrations of bilirubin are with no doubt associated with higher risk of many civilization diseases, while each micromolar increase in bilirubin concentration above the threshold of approximately 10 umol/L is associated with decreased risk. We have described these associations with a detailed discussion in our recent paper (see Vitek L. Bilirubin as a predictor of diseases of civilization. Is it time to establish decision limits for serum bilirubin concentrations? Arch Biochem Biophys 2019), in which we call for re-assessment of current decision limits for serum/plasma bilirubin concentrations.
Based on studies published so far (many of them by us), it is highly likely that low serum bilirubin concentrations are due to increased consumption of this antioxidant and anti-inflammatory substance under conditions of inflammation-induced oxidative stress. See also the addition in the Discussion Section (p. 11), and Conclusion (p. 13).
- In the TAS measurement, how much of the antioxidant capacity is from bilirubin? If substantial, this could explain the correlation. This needs to be discussed.
Reply: The contribution of bilirubin to TAS seems to be quite high. As can be implied from Tab. 5a, the difference in TAS between the groups of quartile 1 and 4 bilirubin concentration is approximately 80 mmol/L (1.98 - 1.90 mmol/L), while the difference between the median values of bilirubin concentrations in the individual quartiles is approximately 31 mmol/L indicating twice as high an increase in TAS as could be expected based on molarity – most likely due to the biliverdin-bilirubin amplification cycle (see Sedlak and Snyder. Bilirubin benefits: cellular protection by a biliverdin reductase antioxidant cycle. Pediatrics 2004). These data are in good agreement with previous results published in our study in 2002, in which we used commercial serum with defined TAS (Randox Lab) and added bilirubin to increase its concentration in the serum by 6.25, 12.5, 25, 50 and 100 mmol/L. Also in these studies, a similar and much higher pattern of increased TAS was demonstrated (see Vitek et al. Gilbert syndrome and ischemic heart disease: a protective effect of elevated bilirubin levels. Atherosclerosis 2002). Appropriate discussion on this important topic was added into Discussion Section (see p. 11, and also our Reply to Comment 1 of Reviewer 1).
- Also, please provide more information on exactly what their TAS assay kit measures.
Reply: The principle of the assay has been provided in full detail in the MM Section (see p. 3).
- Please provide post hoc test used for all ANOVAs.
Reply: ANOVA on Ranks with Dunn´s post-hoc analyses were used for all pairwise comparisons. This information is now stated in the Legend of the Tables 1,2 (omitted in the original Ms.), and 5-8.
- Please discuss possible reasons for differences in males vs. females.
Reply: Discussion on the possible reasons for differences between males and females has been added into the Discussion Section (see p. 13).
- The discussion focusses largely on CRP with relatively little on mechanisms related to bilirubin. Especially for a special topic on heme oxygenase, more focus on bilirubin metabolism is in order.
Reply: Discussion on possible anti-inflammatory effects of bilirubin has been added into the Discussion Section (see p. 12).
Round 2
Reviewer 1 Report
While the description of the TAS method has been expanded, no additional data has been provided to validate TAS results. Limitations of the TAS method are neither stated. Down this line, the oxidative stress argument has not been lessened.
Author Response
We agree that the study has a limitation in use of TAS as a single analytical method to assess the oxidative stress defense status. It is known that this test not always reflects the oxidative stress status completely. Antioxidant defense system is very complex, and the use of a single method analyzing presence of antioxidants in the biological material, such as TAS may be insensitive to subtle changes that can cause oxidation–antioxidant system imbalance. It is thus recommended to assess both antioxidant and oxidant status simultaneously with calculation of specific oxidative stress indexes covering a number of factors involved, which appear to be more reliable to assess a dynamic equilibrium between oxidants and antioxidants in the biological system.
(Feng et al. Analysis of the diagnostic efficiency of serum oxidative stress parameters in patients with breast cancer at various clinical stages. Clin Biochem 2016;49:692-8; Wu et al. Significance of Serum Total Oxidant/Antioxidant Status in Patients with Colorectal Cancer. PLoSOne 2017;12: e0170003;
Veglia F, et al. OXY-SCORE: a global index to improve evaluation of oxidative stress by combining pro- and antioxidant markers. Methods Mol Biol 2010;594:197-213.).
Detailed discussion on these limitations of TAS in assessment of oxidative stress defense system status has been added, including the reasoning for use of this parameter in our study (p. 11-12).